# Segmentectomy for ground glass-dominant invasive lung cancer with tumour diameter of 2–3 cm: protocol for a single-arm, multicentre, phase III trial (ECTOP1012)

Shiqi Chen  ,[1,2,3] Qingyuan Huang,[1,2,3] Fangqiu Fu,[1,2,3] Zezhou Wang,[3,4] Yang Zhang,[1,2,3] Haiquan Chen[1,2,3]

SC, QH and FF contributed equally.

YZ and HC are joint senior authors.

**Correspondence to**
Haiquan Chen;
hqchen1@yahoo.com

## ABSTRACT

**Introduction** Previous studies demonstrated that wedge resection is sufficient for ground glass-dominant lung adenocarcinoma (LUAD) with tumour diameter ≤2 cm, however, the optimal surgical type for ground glass-dominant LUAD with tumour diameter of 2–3 cm remains unclear. The purpose of this trial is to investigate the safety and efficacy of segmentectomy for ground glass-dominant invasive LUAD with tumour size of 2–3 cm.

**Methods and analysis** We initiated a phase III trial to investigate whether segmentectomy is suitable for ground glass-dominant invasive LUAD with tumour size of 2–3 cm. This trial plans to enrol 307 patients from multiple institutions including four general hospitals and two specialty cancer hospitals over a period of 5 years. The primary endpoint is 5 year disease-free survival. Secondary endpoints are lung function, 5 year overall survival, the site of tumour recurrence and metastasis, segmentectomy completion rate, radical segmentectomy (R0 resection) completion rate and surgery-related complications.

**Ethics and dissemination** This trial has been approved by the Ethics Committee of Fudan University Shanghai Cancer Centre (reference 2212267-18) and by the institutional review boards of each participating centre. Written informed consent is required from all participants. The study results will be published in a peer-reviewed international journal.

**Trial registration number** NCT05717803.

## INTRODUCTION

Lung cancer is the most commonly diagnosed malignancy in men and the leading cause of cancer-associated mortality in both men and women.[1] Since the national lung screening trial demonstrated that screening with low-dose CT (LDCT) could increase the detection of lung cancer, and reduce the risk of mortality from lung cancer, LDCT is widely used in the screening for lung cancer.[2] LDCT screening leads to a dramatic increase in the detection of small-sized lung nodules, especially ground-glass opacity (GGO).[3] The shift

---

**STRENGTHS AND LIMITATIONS OF THIS STUDY**

⇒ The trial is conducted in multiple centres, increasing the generalisability of the results.
⇒ Both intraoperative frozen sections and postoperative paraffin sections will be evaluated by two experienced pathologists, making the pathologic diagnosis reliable.
⇒ A key limitation of the trial is lack of a comparator group; considering the indolent nature and excellent prognosis of ground glass-dominant lung adenocarcinoma, lobectomy might be too invasive, so a control-arm of lobectomy is not set.

---

of the spectrum of radiological characteristics challenges lobectomy as the 'gold standard' of surgical treatment for early-stage lung cancer, which was established in the 1990s by the Lung Cancer Study Group trial.[4]

From 2002 to 2004, the JCOG0201 trial included patients with clinical T1N0M0 (≤3 cm in tumour diameter) peripheral lung cancer, and defined radiological non-invasive lung adenocarcinoma (LUAD) as tumour size ≤2 cm with consolidation-to-tumour ratio (CTR) ≤0.25.[5] Afterward, the JCOG1211 trial found that the 5 year recurrence-free survival (RFS) of patients undergoing segmentectomy for the 'radiologically invasive' tumours (tumour diameter ≤2 cm with CTR between 0.25 and 0.50, or tumour diameter between 2 cm and 3 cm with CTR≤0.50) is 98.0%, exceeding the 87% of the preset threshold.[6] However, this trial was criticised on the basis that about half of the included patients were pathologically confirmed as adenocarcinoma in situ (AIS) and microinvasive adenocarcinoma (MIA). For these patients (AIS/MIA), wedge resection is sufficient, and segmentectomy may be unnecessary. Additionally, this also limits the statistical power to

demonstrate the efficacy of pathologically invasive LUAD. Recently, our team performed a retrospective study and found that wedge resection for peripheral invasive LUAD with tumour diameter ≤2 cm and CTR≤0.5 could lead to excellent long-term survival.[7]

This single-arm, multicentre, phase III trial (ECTOP1012) aims to investigate whether segmentectomy is suitable for ground glass-dominant invasive lung cancer with tumour size of 2–3 cm.

## METHODS AND ANALYSIS
### Study design
ECTOP1012 is a single-arm, multicentre, confirmatory phase III trial to evaluate the safety and efficacy of segmentectomy for ground glass-dominant invasive lung cancer with size of 2–3 cm. Four general hospitals and two specialty cancer hospitals will participate in this trial. As the prognosis of these patients is fairly excellent, a large sample size would be required if a randomised controlled trial (RCT) is to be performed, so this study is designed as a single-arm confirmatory trial. The tumour size should be between 2 and 3 cm, and the CTR should not be greater than 0.5 on preoperative high-resolution CT. The surgical procedure for included patients is segmentectomy, which includes single-segment resection, combined segment resection, trisegmentectomy and basal segment resection. Tumours located in the right middle lobe are excluded in this trial (figure 1).

The trial started in February 2023; the estimated primary completion date is December 2028 and the estimated study completion date is December 2029. We used the Standard Protocol Items: Recommendations for Interventional Trials (SPIRIT) checklist when preparing this report.[8]

### Objectives
The primary objective of ECTOP1012 is to assess the efficacy of segmentectomy for ground glass-dominant invasive lung cancer with size of 2–3 cm.
The secondary objectives include:
1. Surgical safety of segmentectomy for ground glass-dominant invasive lung cancer with size of 2–3 cm.
2. Loss of lung function after segmentectomy.
3. Tumour recurrence rate after segmentectomy.

### Endpoints
The primary endpoint is 5 year disease-free survival (5 year DFS). DFS events are defined as tumour recurrence due to lung cancer or death from any cause. Alive patients

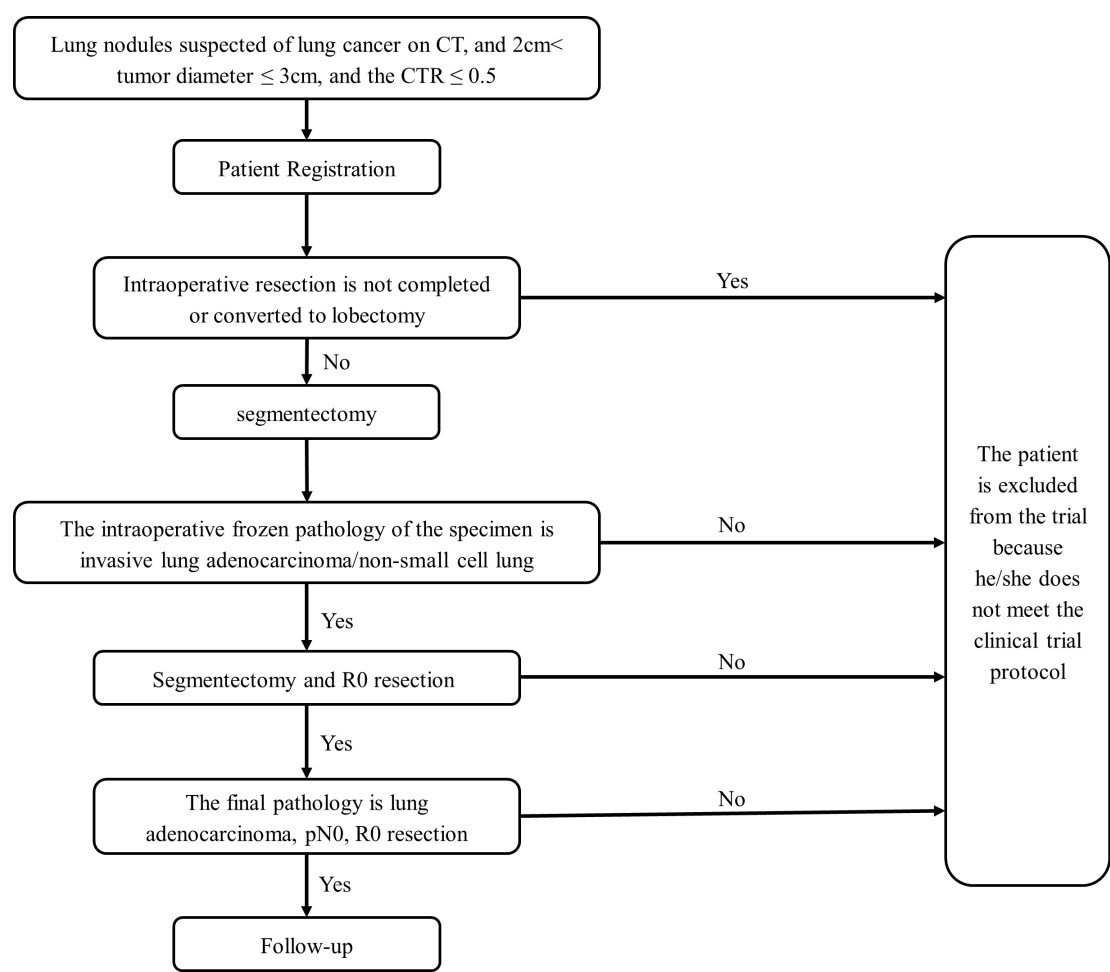

**Figure 1** Study flowchart. CTR, consolidation-to-tumour ratio.

without recurrence at the last follow-up are defined as censored.

The secondary endpoints are lung function, 5 year overall survival, the site of tumour recurrence and metastasis, segmentectomy completion rate, radical segmentectomy (R0 resection) completion rate and surgery-related complications.

### Eligibility criteria
#### Inclusion criteria
All enrolled patients should meet the following criteria:
1. Written informed consent.
2. 18–80 years old (including cut-off value) at the time of signing the informed consent.
3. Eastern Cooperative Oncology Group (ECOG) score of 0 or 1.
4. No prior lung cancer surgery.
5. Invasive LUAD confirmed by intraoperative or postoperative pathological evaluation.
6. A single pulmonary nodule with a predominantly ground-glass component or pure ground-glass on CT, or multiple pulmonary nodules but the main lesion is the above-mentioned nodule.
7. The CTR of the nodule ranges from 0 to 0.5, and the total size is between 2 and 3 cm.
8. No evidence of nodal or distant metastasis.
9. The tumour is evaluated to be completely removed via segmentectomy by thoracic surgeon.
10. No prior radiotherapy or chemotherapy.
   Exclusion criteria

#### Exclusion criteria
Patients are excluded if they meet any of the following criteria:
   1.CTR is not between 0 and 0.5 or the total nodule size is not between 2 and 3 cm.
   2.The lesion cannot be completely surgically removed by segmentectomy.
   3.Lung malignancies other than lung adenocarcinoma confirmed by cytological or histopathological evaluation.
   4.Prior lung cancer surgery.
   5.Non-treatment-naïve patients.

### Withdrawal or termination criteria
All data in this trial are obtained based on existing examinations and procedures and the trial does not cause any additional harm to patients. The specifics of the patient's withdrawal from this study are as follows:
1. The patient's data shall not be used for the clinical study.
2. Patient enrolment error (eg, patient does not meet enrolment criteria).
3. Patients voluntarily propose to withdraw from the trial.

### Treatment
#### Segmentectomy and lymph node dissection
Pulmonary GGOs that are evaluated to be removed via segmentectomy by thoracic surgeons could be included in the study. Anatomical segmentectomy could be single segmentectomy (eg, S3 resection), combined segmentectomy (eg, S7+S8 resection), basal segmentectomy (eg, S7–S10 resection) and trisegmentectomy (eg, left S1–S3 resection). In addition, the use of wedge resection to manage the patient's other non-major lesions is allowed in this study.

Based on the results of previous studies, there was no lymph node metastasis in lung cancer with a CTR<0.5,[9] so there is no further provision for mediastinal lymph node dissection in this study, and the surgeon could choose the extent of mediastinal lymph node dissection according to personal experience. The lymph nodes exposed during the operation need to be removed and sent for pathological examination.

### Procedures
#### Preoperative CT imaging evaluation and clinical staging
All patients are required to undergo a thin-slice (1 mm) contrast-enhanced CT scan of the chest before enrolment. According to the CT scan results, the long diameter of the tumour is the maximum diameter of the tumour (solid component+ground-glass component) on the CT cross-section. The size of the solid component is the maximum diameter of the solid component on the CT cross-section. CTR is defined as the ratio of solid component size to tumour diameter. For pure ground-glass nodules in which there are no solid components, the CTR is 0. cN0 is diagnosed when the short axis of lymph nodes of intrapulmonary, interlobar, hilar and mediastinal are shorter than 10 mm on contrast-enhanced CT or no uptake of lymph nodes on PET/CT.[10] The staging in this study is performed using the 8th edition of the TNM lung cancer staging system.[11] Patients will be admitted to the hospital for surgery after the CT or PET-CT is evaluated by the radiologist, and the interval is 1 month.

#### Intraoperative frozen pathological evaluation criteria
All primary tumours are sent to the pathology room for frozen section diagnosis immediately after resection. Frozen pathology is taken from the maximum diameter of the tumour. Diagnosis is based on the 2015 WHO classification criteria.[12] Lesions diagnosed as invasive lung adenocarcinoma/non-small cell lung cancer by intraoperative frozen are included in this study.

#### Postoperative pathological evaluation criteria
The postoperative pathological diagnosis of the tumour using paraffin sections is based on the 2015 version of the WHO classification criteria, and patients with lung adenocarcinoma are included in this study. All pathology reports are evaluated by two experienced pathologists.

 

## Statistical analysis

### Analysis sets

Full analysis set (FAS): determined according to the intention-to-treat principle, including all enrolled cases who have undergone segmentectomy.

Per-protocol set (PPS): subset of the FAS population, including all cases that meet the protocol and have completed case report form.

In this trial, FAS analysis is used for baseline data and validity analysis. The main efficacy indicators are will be analysed in the FAS and PPS data sets. However, the analyses of the FAS data set are considered primary. When analyses of the FAS and PPS reach consistent conclusions, confidence in the conclusions can be increased. For the FAS data set, the missing data after the fall are estimated using the last observation carried forward method.

The laboratory examination data and adverse events will be analysed using SPSS and Stata software.

### Calculation of sample size

According to the results of previous studies, for patients undergoing lobectomy with tumour sizes of 2–3 cm, CTR≤0.5 and the 5 year DFS was 97.0% after surgery.[13] In this trial, we assumed that the expected 5 year DFS for these patients undergoing segmentectomy is 95%. A sample size of 292 achieves 90.193% power to detect a non-inferiority proportion of 90% (5% as the non-inferior boundary) using a one-sided exact test with a target significance level of 0.0250. Considering a possible dropout rate of 5%, a total of 307 cases need to be included in this trial.

### Data access

The Data Management Coordinating Centre will oversee the intrastudy data-sharing process, with input from the Data Management Subcommittee. All Principal Investigators will be given access to the cleaned data sets. Project data sets will be housed on the Project Accept website and/or the file transfer protocol site created for the study, and all data sets will be password-protected. Project Principal Investigators will have direct access to their own site's data sets and will have access to other sites' data by request. To ensure confidentiality, data dispersed to project team members will be blinded to any identifying participant information.

### Patient and public involvement

None.

## ETHICS AND DISSEMINATION

The protocol and informed consent form have been reviewed and approved by the Ethics Committee of Fudan University Shanghai Cancer Centre (reference 2212267-18) and by the institutional review boards of each participating centre. Only patients who can fully understand the risks, benefits and potential adverse events of this trial will be enrolled after voluntarily signing an informed consent form (online supplemental Material 1). Informed consent should be in line with the Declaration of Helsinki.

The personal data collected and processed by the patients enrolled in this study will be limited to those necessary to complete the purposes of this study.

The results of the trial will be updated on ClinicalTrials. gov, published in peer-reviewed journals and presented at international academic conferences.

**Author affiliations**
[1]Departments of Thoracic Surgery and State Key Laboratory of Genetic Engineering, Fudan University Shanghai Cancer Center, Shanghai, China
[2]Institute of Thoracic Oncology, Fudan University, Shanghai, China
[3]Department of Oncology, Shanghai Medical College, Fudan University, Shanghai, China
[4]Department of Cancer Prevention, Fudan University Shanghai Cancer Center, Shanghai, China

**Contributors** HC and YZ: conceiving the study. SC, QH and FF: drafting the protocol. QH and FF: registering the protocol and ethical submission. QH and HC: funding acquisition. YZ and HC: supervision and revising the draft. ZW: calculation of sample size and statistical analysis interpretation. All authors contributed to the article and approved the submitted version. HC is the guarantor.

**Funding** This work was supported by the National Natural Science Foundation of China (81930073, 82203504), the Shanghai Technology Innovation Action Project (20JC1417200), the Cooperation Project of Conquering Major Diseases in Xuhui District (XHLHGG202101) and the National Key R&D Programme of China (2022YFA1103900).

**Competing interests** None declared.

**Patient and public involvement** Patients and/or the public were not involved in the design, or conduct, or reporting or dissemination plans of this research.

**Patient consent for publication** Not applicable.

**Provenance and peer review** Not commissioned; externally peer reviewed.

**ORCID iD**
Shiqi Chen http://orcid.org/0000-0002-3543-4939

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
