## [Reviewer comments · BMJ Open]

ARTICLE DETAILS

TITLE (PROVISIONAL)	Segmentectomy for ground glass-dominant invasive lung cancer with tumour diameter of 2-3 cm: protocol for a single-arm, multicentre, phase III trial (ECTOP1012)
AUTHORS	Chen, Shiqi; Huang, Qingyuan; Fu, Fangqiu; Wang, Zezhou; Zhang, Yang; Chen, Haiquan

VERSION 1 – REVIEW

REVIEWER	Gonzalez, Michel Centre Hospitalier Universitaire Vaudois-1011, Thoracic Surgery
REVIEW RETURNED	25-Apr-2024

GENERAL COMMENTS	Dear authors, thank you for your protocol that addresses an important question on management of GGO predominant invasive adenocarcinoma. Some remarks: - I understand that lymph node dissection in this situation is probably virtually negative. But, in my opinion it should be performed, because you will exclude patients with N1 disease in your algorithm. So, if there is no lymph node dissection this will probably make an underestimation of lymph node involvement and potentially include some patients with N+ disease.- You should mention in the method the accepted interval between CT or PET-CT and surgical operation.- In term of surgical approach, all approaches are allowed (VATS, RATS, open thoracotomy)? Otherwise, congratulations for your effort.
---

REVIEWER	Yang, Chi-Fu Jeffrey Massachusetts General Hospital - Harvard Medical School Center for Nervous System Repair
REVIEW RETURNED	15-May-2024

GENERAL COMMENTS	The authors should be congratulated for launching their study to evaluate segmentectomy for GGO's that are 2-3 cm. The study results will certainly be impactful and be widely anticipated by thoracic surgeons. I have some suggestions:
---

	1) Consider adding a citation for this sentence: LDCT screening leads to a dramatic increase in the detection of small-sized lung nodules, especially ground-glass opacity (GGO). 2) For the following sentence, can the authors provide some numbers for the long-term survival rates? "Recently, our team performed a retrospective study and found that wedge resection for peripheral invasive LUAD with tumor diameters ≤ 2 cm and CTR ≤ 0.5 could lead to excellent long-term survival." 3) Can the authors provide updated citations -- for example, JCOG1211 has already been published and the authors should use this citation instead of the old one that they provided: Lancet Respir Med 2023; 11: 540–49
--	---

VERSION 1 – AUTHOR RESPONSE

Reviewer: 1

Dr. Michel Gonzalez, Centre Hospitalier Universitaire Vaudois-1011

Comments to the Author:

Dear authors, thank you for your protocol that addresses an important question on management of GGO predominant invasive adenocarcinoma.

Some remarks:

- I understand that lymph node dissection in this situation is probably virtually negative. But, in my opinion it should be performed, because you will exclude patients with N1 disease in your algorithm. So, if there is no lymph node dissection this will probably make an underestimation of lymph node involvement and potentially include some patients with N+ disease.

Response: Thank you for your insights. We totally understand your concern, which seems reasonable. Actually, we had conducted a retrospective study to explore whether it's necessary for these patients to perform lymph node dissection, and we found that there was no lymph node metastasis in lung cancer with a CTR < 0.5 (**PMID: 31863756**). Therefore, we continued a prospective study (**PMID: 36841542**) to further validate our finding in the retrospective study, and we got the same conclusion. Therefore, lymph node dissection is not mandatory in our protocol, but the surgeon can choose whether to dissect the lymph nodes based on personal experience, and the lymph nodes exposed during the operation need to be removed and sent for pathological examination.

- You should mention in the method the accepted interval between CT or PET-CT and surgical operation.

Response: The interval is one month, and this has been added in the manuscript (**Page7, Line185-186**).

- In term of surgical approach, all approaches are allowed (VATS, RATS, open thoracotomy)?

Response: Yes, all approaches are allowed (VATS, RATS, open thoracotomy). Surgeons could choose an appropriate surgical approach (VATS or RATS or open thoracotomy) which is suitable for the patients according to the patients' general condition, radiologic characteristics of the tumor, personal wishes and the surgeons' specialty and preference.

Otherwise, congratulations for your effort.

Response: Thank you for your comments and encourage.

Reviewer: 2

Dr. Chi-Fu Jeffrey Yang, Massachusetts General Hospital - Harvard Medical School Center for Nervous System Repair

Comments to the Author:

The authors should be congratulated for launching their study to evaluate segmentectomy for GGO's that are 2-3 cm. The study results will certainly be impactful and be widely anticipated by thoracic surgeons.

I have some suggestions:

1) Consider adding a citation for this sentence: LDCT screening leads to a dramatic increase in the detection of small-sized lung nodules, especially ground-glass opacity (GGO).

Response: Thanks for your comments. We have added a citation for this sentence (**Page3, Line71, Ref. 3**).

2) For the following sentence, can the authors provide some numbers for the long-term survival rates?
"Recently, our team performed a retrospective study and found that wedge resection for peripheral invasive LUAD with tumor diameters ≤ 2 cm and $CTR \leq 0.5$ could lead to excellent long-term survival."

Response: We appreciate your comment. This is a retrospective study recently published in J Thorac Cardiovasc Surg (2024 Mar;167(3):797-809, PMID: 37385528). The 5-year recurrence-free survival (RFS) of patients with LUAD ≤ 2 cm and $0.25 < CTR \leq 0.5$ receiving wedge resection was 96.89%, and the 5-year RFS of patients with LUAD ≤ 2 cm and $CTR \leq 0.25$ receiving wedge resection was 100%. The 5-year lung cancer-specific overall survival (LCS-OS) of patients with LUAD ≤ 2 cm and $0.25 < CTR \leq 0.5$ receiving wedge resection was 97.87%, and the 5-year LCS-OS of patients with LUAD ≤ 2 cm and $CTR \leq 0.25$ receiving wedge resection was 100% (See Ref. 7 for more details).

3) Can the authors provide updated citations -- for example, JCOG1211 has already been published and the authors should use this citation instead of the old one that they provided: Lancet Respir Med 2023; 11: 540–49

Response: We have thoroughly updated the citations, including the one you mentioned (Page3, Line81, Ref. 6).

Additionally, we have added one new author (Dr. Zezhou Wang, Department of Cancer Prevention, Fudan University Shanghai Cancer Center) who was inadvertently excluded during initial submission. Dr. Wang contributed to the calculation of sample size and the statistical analysis interpretation. Besides, we added a citation for 2015 WHO classification criteria of lung tumors (Page7, Line190, Ref. 12) and a citation for Eighth edition of the TNM classification for lung cancer (Page7, Line185, Ref. 11).

VERSION 2 – REVIEW

REVIEWER	Gonzalez, Michel Centre Hospitalier Universitaire Vaudois-1011, Thoracic Surgery
REVIEW RETURNED	13-Jun-2024
GENERAL COMMENTS	Dear authors, you addressed all questions.